# Exploring the Potential of Multi-Temporal Crop Canopy Models and Vegetation Indices from Pleiades Imagery for Yield Estimation

**Dimo Dimov [1] and Patrick Noack [2,*]**

1   Geocledian GmbH, 84028 Landshut, Germany
2   Competence Center for Digital Agriculture, University of Applied Sciences Weihenstephan-Triesdorf, 85354 Freising, Germany
*   Correspondence: patrick.noack@hswt.de

**Abstract:** In this paper, we demonstrate the capabilities of Pleiades-1a imagery for very high resolution (VHR) crop yield estimation by utilizing the predictor variables from the horizontal-spectral information, through Normalized Difference Vegetation Indices (NDVI), and the vertical-volumetric crop characteristics, through the derivation of Crop Canopy Models (CCMs), from the stereo imaging capacity of the satellite. CCMs captured by Unmanned Aerial Vehicles are widely used in precision farming applications, but they are not suitable for the mapping of large or inaccessible areas. We further explore the spatiotemporal relationship of the CCMs and the NDVI for five observation dates during the growing season for eight selected crop fields in Germany with harvester-measured ground truth crop yield. Moreover, we explore different CCM normalization methods, as well as linear and non-linear regression algorithms, for the crop yield estimation. Overall, using the Extremely Randomized Trees regression, the combination of CCMs and NDVI achieves an $R^2$ coefficient of determination of 0.92.

**Keywords:** photogrammetry; yield estimation; canopy surface models; precision farming; Pleiades

## 1. Introduction

Agriculture is under increasing pressure to transform and innovate its production processes. The growing world population requires a growing amount of food, feed, and raw materials for industrial processes and energy production [1]. At the same time, reducing inputs such as fertilizer and agrochemicals is vital for reducing the negative impacts on the environment and natural resources, such as groundwater and the biosphere [2].

Finally, yet importantly, the declining number of people involved in agricultural production is challenging. It results in the urgent requirement to make human labor more productive or to replace it. On the other hand, it supports a process described as the erosion of knowledge, meaning that specific knowledge of farm managers on the characteristics of fields on a subfield level, until recently used for optimizing production processes, decreases along with the agricultural workforce.

### 1.1. Precision Agriculture

Precision agriculture is a toolset of digital technologies. It aims at supporting human labor or replacing it. It also provides data and information on the spatial and temporal variability of parameters affecting processes in plant production, including soil maps, nutrient maps, yield maps, data collected with vehicle-mounted sensors, and satellite data [3]. The aims of precision agriculture are (1) raising yield, (2) minimizing input and input costs, and (3) reducing negative impacts on natural resources and the biosphere. Therefore, it helps to address the challenges mentioned above that agriculture is facing, now and in the future.

Yield data are crucial in precision plant production [4]. On the one hand, understanding the spatial variability of yield potential within fields is of substantial importance. It helps to delineate management zones requiring different treatments for tillage, seed density, fertilization, and the application of agrochemicals. On the other hand, yield data help evaluate and validate the effects of site-specific treatment, equivalent to the local adjustment of agricultural tasks, such as tillage, seeding, fertilizing, and the application of agrochemicals.

Yield data are usually acquired during harvest by applying the GNSS and dynamic weighing technology integrated into combine harvesters [5]. However, the adoption of yield data collection systems could be higher. Farmers are unwilling to invest in additional technology with an unknown return on investment, which requires additional attention before and during harvest. Contractors providing harvest as a service for farmers are reluctant to introduce yield data collection technology, as delivering yield data is usually not rewarded. Last, but not least, yield data collected in the field is or may be erroneous, and they therefore require treatment before processing and further use in precision farming applications, such as site-specific treatment. For the reasons mentioned above, developing processes that allow estimating local yield on fields without applying yield collection technology is more than desirable.

### 1.2. Crop Canopy Models (CCM)

Photogrammetric methods applied to imagery from UAVs and other airborne sensors have been widely used for precision farming, forestry, and hydrological (e.g., irrigation engineering, glaciology) applications. High-resolution imagery (with grid sampling sizes of less than 30 cm) offers high-precision three-dimensional modeling and object height estimation (crop height, stem height, tree canopy height). Kümmerer et al. have lately shown that Crop Canopy Models (CCMs) are very suitable for assessing the yield of mixed cover crop stands [6]. Bendig et al. has shown that a combination of spectral measurements and the determination of the canopy helps to estimate barley biomass [7].

Usually, laser scanning (light detection and ranging, LiDAR) data are widely used to generate very high-resolution CCMs and three-dimensional object extraction, but they are spatially limited and costly [8,9].

Satellite technologies are yet limited to coarser spatial resolution, especially for creating topographic Digital Surface Models (DSM), with a spatial resolution of 1 m, at best. As in any photogrammetric model, the vertical accuracy depends on the density of observations and sufficient incident angles, the provision of GNSS-based Ground Control Points (GCPs), and other available geodetic data.

Such DSMs can be generated by high-resolution Pleiades 1a/1b or SPOT-6/-7 imagery, as both satellite platforms provide stereo imaging. They are used for the estimation of building volumes and building heights, as shown in [10,11]. Forestry applications are rare, but good results were achieved for deriving forest metrics in the alpine regions [12]. Three-dimensional information from stereo imaging was used as an additional input feature to enhance the mapping accuracy for land cover and land use classifications [13]. New research includes generative models, such as the ETH Global Canopy Height 2020 product [14], based on a probabilistic deep learning model from Sentinel-2 imagery.

One clear advantage of satellite data versus the application of UAV and aerial imagery is the capability to observe almost any larger area worldwide, especially in inaccessible areas, avoiding costly field campaigns and areas with legal restrictions on UAV operations. However, such remote monitoring can only sometimes rely on the setup of geodetically verified GCPs, leading to accuracy challenges in photogrammetric modeling.

### 1.3. Yield Estimation Based on Satellite Data

Refs. [15,16] provide a broad overview of applying machine learning methods for yield estimation based on satellite data. However, most of the approaches are limited to the application of satellite imagery. Approaches to apply DSM, such as those provided by the Pleiades 1a/1b system, should be more served. Ref. [17] reports that modeling yield

from satellite imagery results in low correlations (mean r = 0.12) between the actual yield recorded on a combine harvester and the modeling results. The author investigated data from different sensors and derived different indices collected over a period of 13 years from more than 900 fields with cereals and canola crops. Ref. [18] reports that recurrent neuronal networks helped to estimate yield with high accuracy (max. $R^2$: 0.88) when taking an image-based approach. The performance decreased when predicting data from unseen years, especially those with weather conditions not reflected in the training data sets.

The overarching goal of this study is to explore the feasibility of satellite-based predictor variables derived from the horizontal-spectral information, through Normalized Difference Vegetation Indices (NDVI), and the vertical-volumetric crop characteristics, through CCMs, from Pleiades-1a imagery.

## 2. Materials and Methods

### 2.1. Study Area and Data Collection

The study area is located in the municipality of Weidenbach, district of Ansbach, Germany (49°12′17.30″N, 10°39′42.10″E), with an average elevation above sea level of 440 m. The predominant soil is a sandy loam. The Stagnic-Cambisols developed from shallow loess over in situ weathered sandstone–claystone (Keuper). The long-term average annual temperature in the study region is 8.9 °C, and the average annual temperature in 2020 was 9.7 °C. The long-term average annual precipitation is 650 mm. Precipitation fell far below this level in 2020 (545 mm) and the two preceding years (561 mm in 2019 and 566 m in 2018), producing a massive deficit in the water balance (−400 mm) [19].

Yield data were collected with a Claas Tucano combine harvester (serial number 83801623) on eight fields (Figure 1), with an area ranging between 1.08 and 4.53 hectares, between 23 July 2020 and 11 August 2020. The crops cultivated were predominantly small grains (winter wheat, winter barley, triticale, and oat), including one field with canola.

To remove erroneous measurements (e.g., due to lodging and resulting speed changes), raw data were filtered on yield measurements (between 2 t/ha and 15 t/ha) as well as forward speed (above 1 km/h) with the software (QGIS Development Team 2022). The spatial resolution of the yield data grids is 5.4 m.

In order to ensure a common reference system and cartographic projection, the yield data have been geodetically transformed from ETRS89 to the coordinate system of the Pleiades data, which is WGS84 UTM Zone 15 North (EPSG 32615). We have applied bilinear interpolation for the pixel resampling to minimize yield value and resolution changes. Table 1 shows the average yield for each of the test fields with the respective crop cultivated: (a) winter barley, (b) winter wheat, (c) winter oats, (d) triticale, and (e) canola.

**Table 1.** Mean yield measurements for each field.

| Field | Crop | Area in ha | Average Yield in t/ha |
|---|---|---|---|
| Rotunderacker | Winter barley | 3.86 | 7.19 |
| Sichertsbuehl | Canola | 2.86 | 4.79 |
| Spessartacker | Winter wheat | 2.99 | 6.86 |
| Steinbruch | Winter barley | 2.92 | 6.13 |
| Theateracker | Winter oats | 1.08 | 6.87 |
| Weihenacker | Winter barley | 1.87 | 6.15 |
| Wosanger | Winter wheat | 3.98 | 5.74 |
| Ziegelacker | Triticale | 4.53 | 7.81 |

The yield value distribution per crop type is shown in Figure 2, with the median values of the three cereal crops being very similar. In contrast, the median values of triticale and canola crops are higher and lower, respectively.

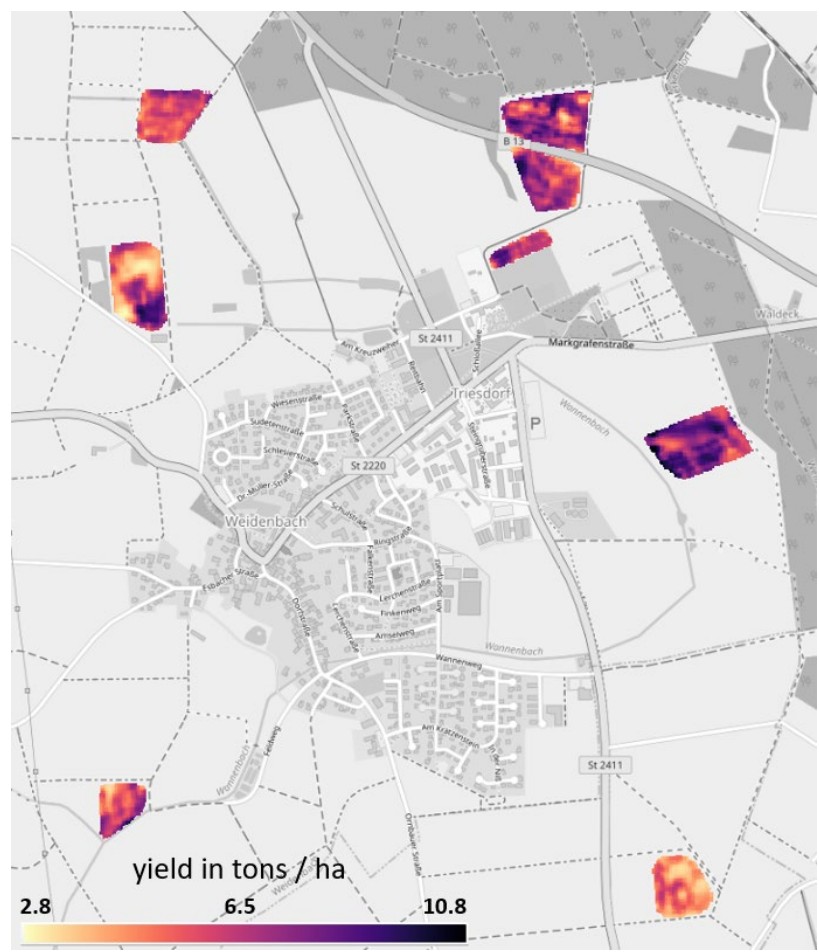

**Figure 1.** Overview map of the research area in Triesdorf, depicting the eight fields with color-coded yield in t/ha (Basemap: OpenStreetMap).

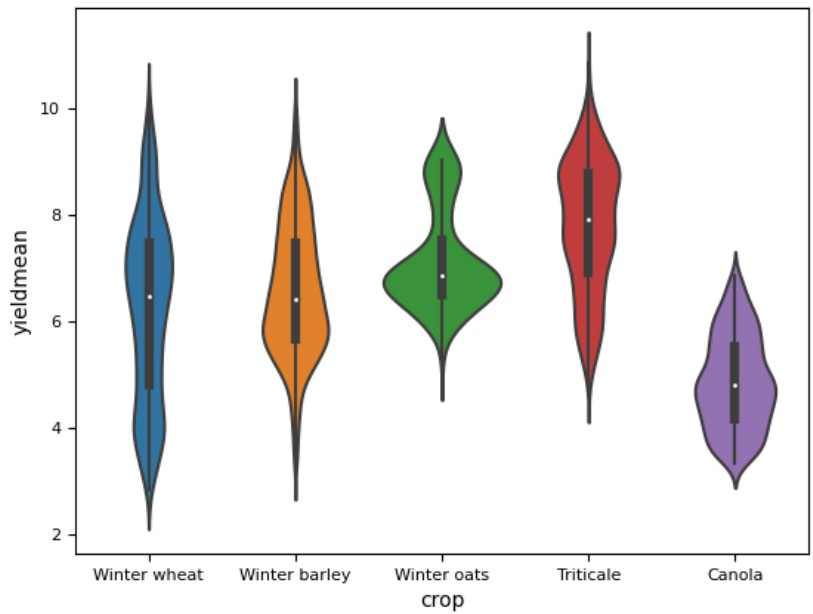

**Figure 2.** Distribution plot of the yield in t/ha for the different crops.

*2.2. Pleiades 1a/1b Data*

We have obtained free-for-research Pleiades 1a/1b bi-stereo data from Airbus Imaging (OneAtlas product) for an area of interest (AOI) of 25 km$^2$ and five different observation dates: 12 June 2020, 26 June 2020, 22 July 2020, 10 August 2020, and 7 September 2020.

The satellite images were selected accordingly with the Pleiades mission's availability and the absence of clouds over the requested AOI. The sensors of the two Pleiades-1a/b satellites, operated by the French National Centre for Space Studies (CNES) and manufactured by EADS, sense the earth's surface with four multispectral channels in 2 m spatial resolution and one panchromatic channel with 0.5 m spatial resolution. They acquire images in agile stereo mode with one camera to observe the same area at different angles and form a stereo image pair to obtain stereo information. This mode is typically used to acquire two or more times the number of observations of the same area at different angles, using the attitude maneuver of the satellite pitch or roll axis. Relying on high satellite agility, the agile stereo mode can make rapid stereo observations of a large area.

Due to the relatively high cost of Pleiades data (in comparison to freely available Landsat or Copernicus data), they has not been extensively used in precision farming or farm management applications for crop monitoring or mapping. Several research studies have been carried out, focusing on the derivation of high-resolution parcel boundary detection [20], derivation of crop characteristics [21], and crop type mapping [22].

**3. Data Analysis**

We have conducted a descriptive analysis that aims to explain the relationship between the horizontal-spectral and vertical-volumetric properties of the crops, as well as to introduce the vertical CCM features as additional independent variables for the supervised regression and yield estimation. This assessment aims to evaluate the level of agreement between those datasets and yield data and enable stratified yield modeling on low and high-productivity zones, respectively. At last, we have extracted the aggregated temporal features to reveal the relative temporal differences between the different dates which experience a significant increase or decrease in chlorophyll content (expressed by NDVI values) and plant height (expressed by the CCM values). Data preprocessing has been conducted with ESRI ArcMAP software, and the data analysis with the python-based data science packages scikit-learn, seaborn, and geopandas.

*3.1. Crop Canopy Model Extraction*

For each observation date, we extracted a DSM based on the provided stereo data using the OrthoEngine of the PCI Geomatica 2016 software (Version 2016 2016-08-04) (https://catalyst.earth/). Tie points were automatically selected through Fast Fourier Transform Phase Matching to ensure relative orientation between the images and the Rational Polynomial Coefficients, delivered with the metadata of the Pleiades stereo images, as described in [23]. All tie point pairs with a root mean square error greater than 0.5 were removed from further analysis to reduce the contribution of tie point mismatching. Epipolar image pairs were created as inputs to the automated DSM extraction with a low smoothing filter for gap filling and Wallis filtering for shadow areas (unmapped areas out of sensor sight). In order to derive the actual vegetation heights, we normalized the height information of each DSM by obtaining a reference Digital Elevation Model (DEM) through three different methods, as no GCP or other ground truth reference data are available.

1.  nDSM_1: In PCI Geomatica, each DSM is converted to a bare-earth DEM by masking out pixel groups with abrupt negative and positive slope changes and spatially interpolating the areas in between from surrounding elevation values
2.  nDSM_2: Each DSM is subtracted from an aggregated per-pixel minima DSM from all observation dates, assuming that the vegetation dynamics would reveal bare soil information, which can be used as a reference height for the crop canopy. This method would not work for anthropogenic objects (buildings, bridges and other infrastructure) or tree objects, as their height would be consistent for all observation dates.

3.  nDSM_3: Each DSM is subtracted from a coarse resolution EU-DEM with 25 m spatial resolution.

### *3.2. Yield Estimation*

The combinatorial use of CCM and NDVI gives essential insights into the N-intake of the plants, as well as maturation and harvesting [24]. It can also be used further for the regressive estimation of synthetical CCM from descriptive data—such as the NDVI, in our case—and to associate horizontal greenness and biomass information with vertical and volumetric structures. As both datasets have a different spatial resolution, each yield value cell has been associated with the aggregated median NDVI and median CCM values. This also accounts for noise and erroneous CCM pixel estimates due to incidence angle insufficiency. The mitigation of such problems is suggested via a high angle of incidence [25].

We used and compared two supervised regression algorithms, Linear Regression (based on ordinary least squares loss function) and Extremely Randomized Trees (ERT), an enhanced version of the well-known Random Forest algorithm, to cope with non-linearity in the feature space [26].

We trained different models from the ground truth data and the extracted NDVI and CCM features from the Pleiades imagery. The ERT model is a forest of decision trees that randomly uses different sets of splits, samples, and features to estimate the given ground truth with the lowest possible absolute error, measured internally. By contrast, the linear model builds a linear regression formula based on all given features by using ordinary least squares loss.

To avoid contamination of the feature dataset with outliers (especially noisy values from the DSM-based features), we used the Local Outlier Factor (LOF) to omit data that are located outside the learned distribution [27]. We randomly held back 5% of the in situ data for validation for each experimental setup and used 95% for training. The different setups are defined as follows:

1.  V1: Regression tests for different crop types
2.  V2: Regression tests for different productivity levels
3.  V3: Regression tests for all data (generic model)
4.  V4: Cross-regression test on one field and inference on all other fields (crop-agnostic transfer)
5.  V5: Multi-temporal vs. single observations

In order to avoid the utilization of correlated features, we have applied Principal Component Analysis (PCA) on the combined dataset consisting of NDVI and CCM features. Furthermore, PCA transformation allows the fusion of multiple datasets with different distributions [28].

The validation of the regression models is performed through different accuracy metrics such as the coefficient of determination ($R^2$), which is a widely used metric, showing how well the ground truth data are explained by the model, as well as the root mean square error (RMSE) and the mean absolute error (MAE); both of them represent the deviation from the actual ground truth measurements and the modelled estimates in tons per hectare.

## 4. Results and Discussion

The inclusion of absolute height information from non-normalized, multi-temporal DSMs, additionally to NDVI independent variables, has proven to improve regression model accuracies and to lower the model error significantly for linear models (especially when the features are PCA-transformed) but not for non-linear models (e.g., Random Forest or ERT), where the contribution of additional features leads to a maximum gain of $R^2$ of about 0.05, which is not significant. On the other hand, our experiments showed that including all features improved the estimation scores for crop-specific regression and model transfer (from one crop to another).

### 4.1. Crop Canopy Models

The relative orientation of the two satellite observations for each date achieves a relative height estimation with an RMSE of less than 0.5 m. However, due to missing GCPs and the use of bi-stereo instead of tri-stereo data, or even multi-bundle adjustment, vertical deviations between the observations of quasi-invariant features such as buildings, forests, and roads occur. Figure 3 shows the extracted nDSM and the associated NDVI image.

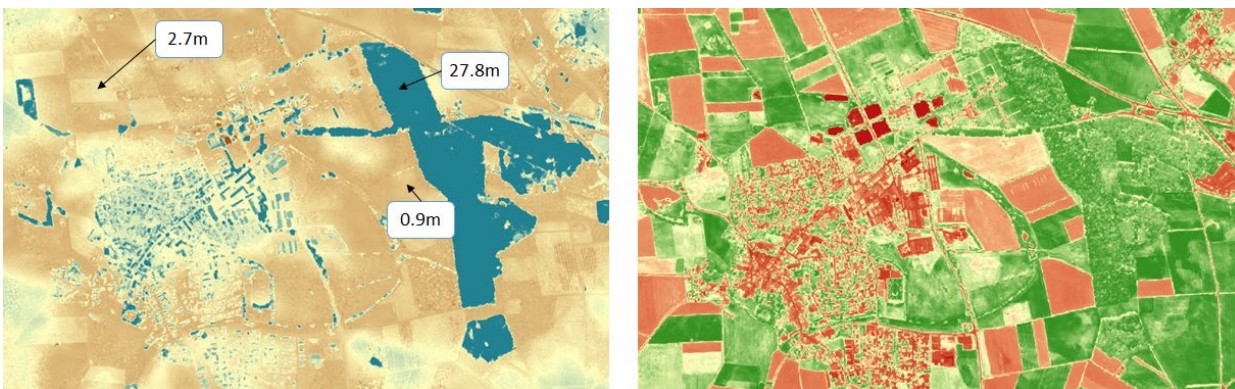

**Figure 3.** Automatically generated nDSM_1 (**left**) and the respective NDVI image (**right**). on 10 August 2020, of the target area.

For better visualization, we have analyzed the standard deviation for 1000 random points between all five observation dates for very low nDSM value ranges (where the nDSM range is less than 1 m height), as shown in Figure 4. We concluded that such average fluctuations of 0.28 m in vertical height are negligible. Based on this analysis, we concluded that the automatically generated nDSM_1 is more suitable than the other two nDSM derivation strategies (with higher fluctuations), and we used it as the final CCM feature dataset.

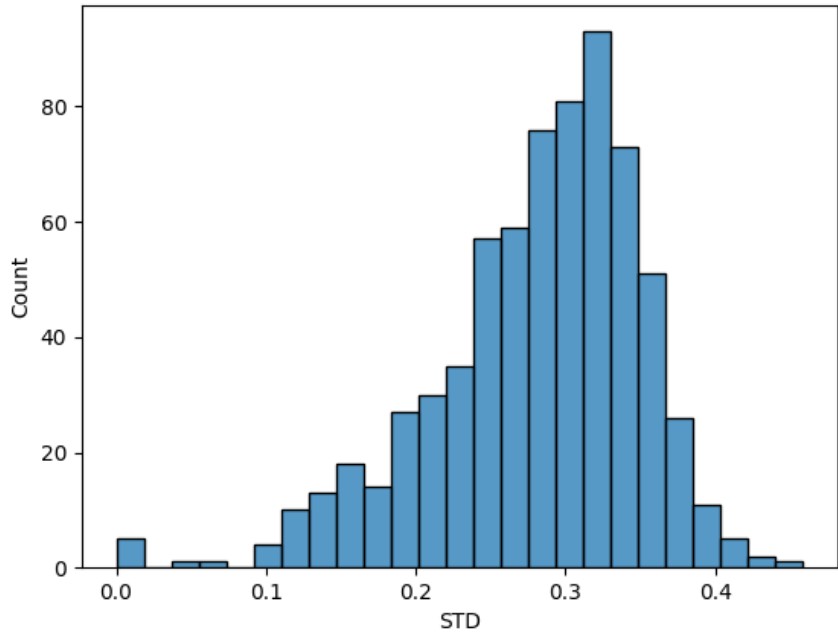

**Figure 4.** Histogram of standard deviations of random pixels with a DSM range between 0 and 1 m between the five different observation dates shows fluctuations in quasi-invariant surface objects, such as infrastructure and forests.

Figure 5 shows three-dimensional views of several parcels where row structures are visible, as well as gaps (sowing/crop failure or other crop damages) and plowing or soil management patterns. Such images could be combined with variable rate application maps for fertilization or other farmer activities.

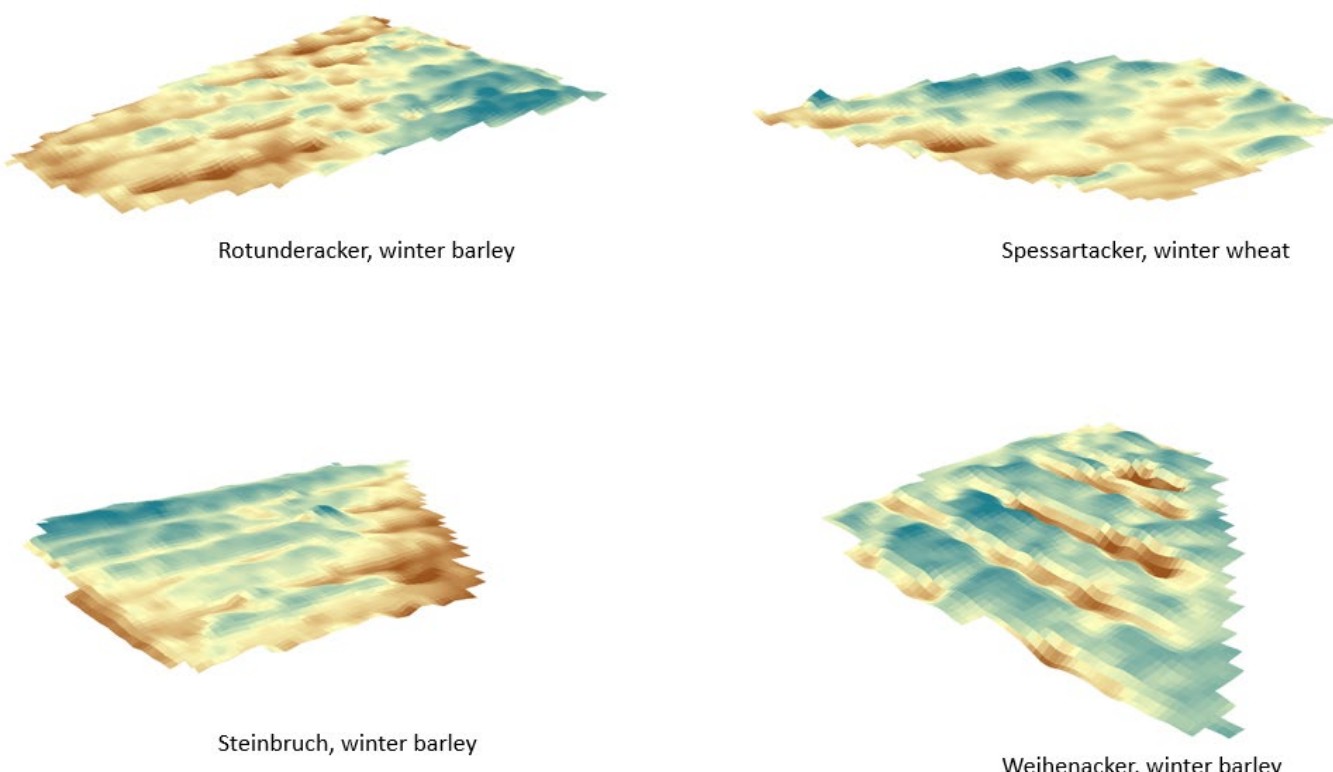

**Figure 5.** CCMs on 26 June for four different fields of winter wheat and winter barley showing different vertical structures (with brown–yellow–blue color-coding representing low–medium–high DSM values, respectively).

### 4.2. Yield Estimation

#### 4.2.1. Per Crop Type

The yield estimation using all CCM and NDVI variables for each crop type led to satisfactory regression results based on the non-linear ERT algorithm, with winter wheat achieving the highest ($R^2$ = 0.92, RMSE = 0.41, MAE = 0.31) and winter oats having the lowest regression accuracy ($R^2$ = 0.74, RMSE = 0.47, MAE = 0.30), as shown in Figure 6. The linear regression, however, failed for the crop types triticale and canola, with $R^2$ less than 0.5, and is not shown in the validation plots. The sampling was performed by drawing model-unseen pixels stratified for a specific crop type using a 95-5 split for training and validation.

The results show that the combined use of all features significantly improves the estimation results for all crop types except for winter oats and wheat, which only achieve a minor improvement. The results with CCM-only features should be considered more for canola.

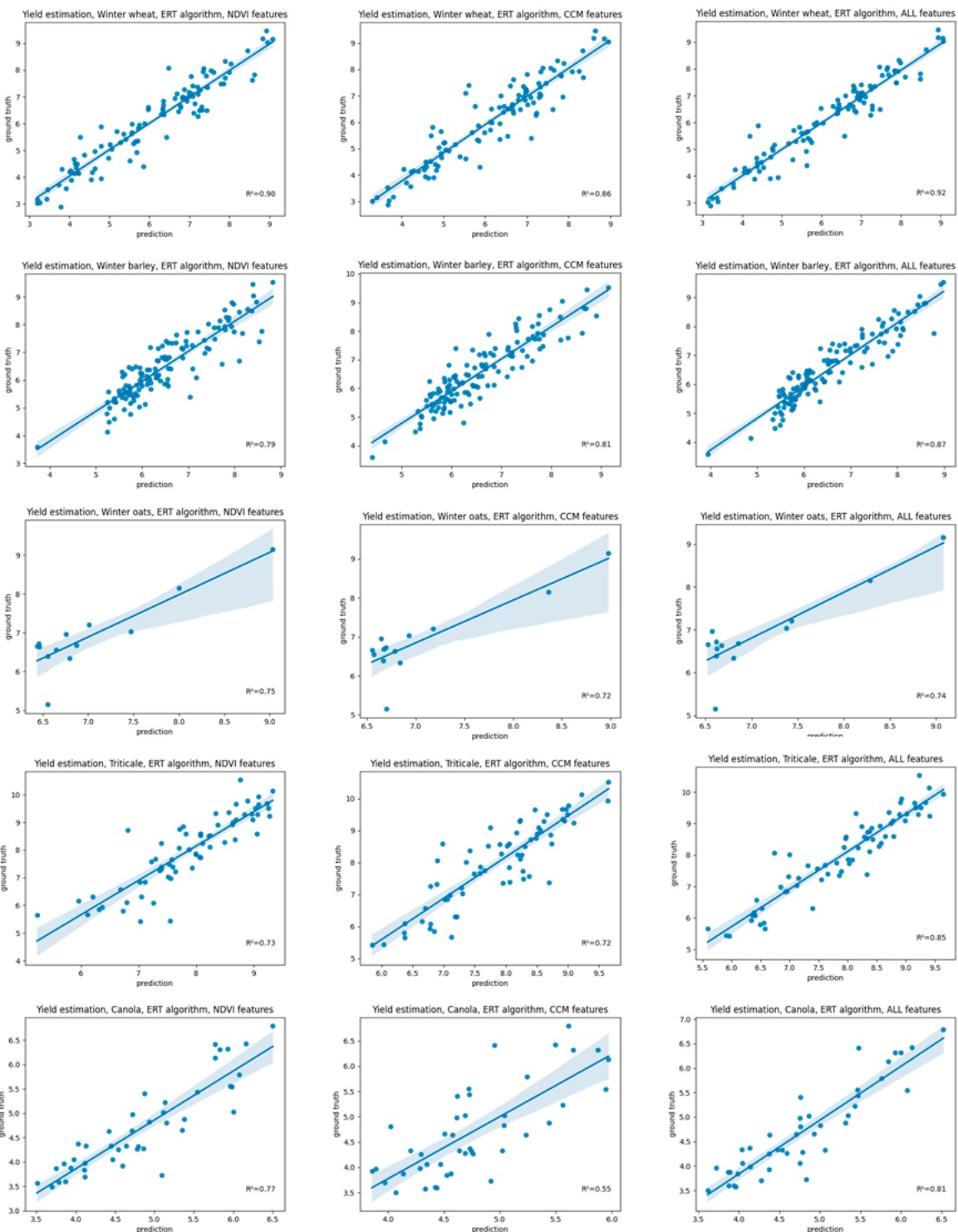

**Figure 6.** Validation results for the yield estimation of 5 different crop types with the ERT algorithm and NDVI features (**left**), CCM features (**middle**), and all features combined (**right**).

### 4.2.2. Per Productivity Level

Another assessment for yield estimation responsiveness of the features was performed by grouping the crop yields in different productivity groups (1 = low, 2 = medium, and 3 = high yield) by statistical thresholding based on the first and third quantiles of the distribution. Results revealed a clear correlation between low yields (productivity group 1) and respective low NDVI and nDSM values across the crops—as shown in Figure 7.

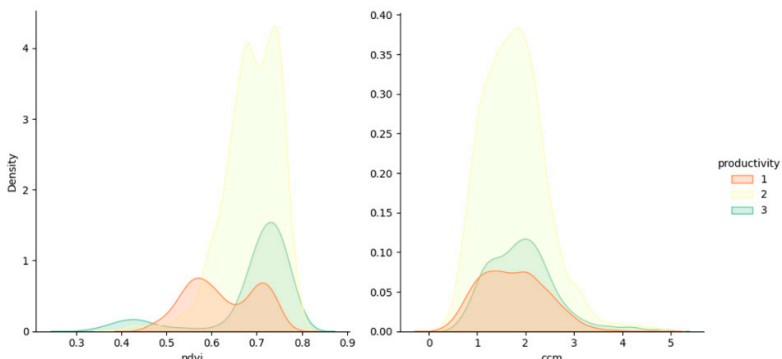

**Figure 7.** Kernel density plots of mean CCM and NDVI features stratified by three productivity levels.

Figure 8 shows different regression results of the linear and Random Forest algorithm for three different productivity levels, taking into account all input features agnostically across all crop types, again proving the assumption that the yield estimation generated better results with a lower error at the lower productivity levels (achieving an $R^2$ score of 0.62, RMSE of 0.35, and MAE of 0.29). In contrast, a distinction of higher yield levels leads to higher estimation errors (with an $R^2$ score of less than 0.4, RMSE of 0.52, and MAE of 0.45) for the ERT and the linear regression algorithms. In contrast, the linear regression only achieves an $R^2$ of less than 0.2 at all levels.

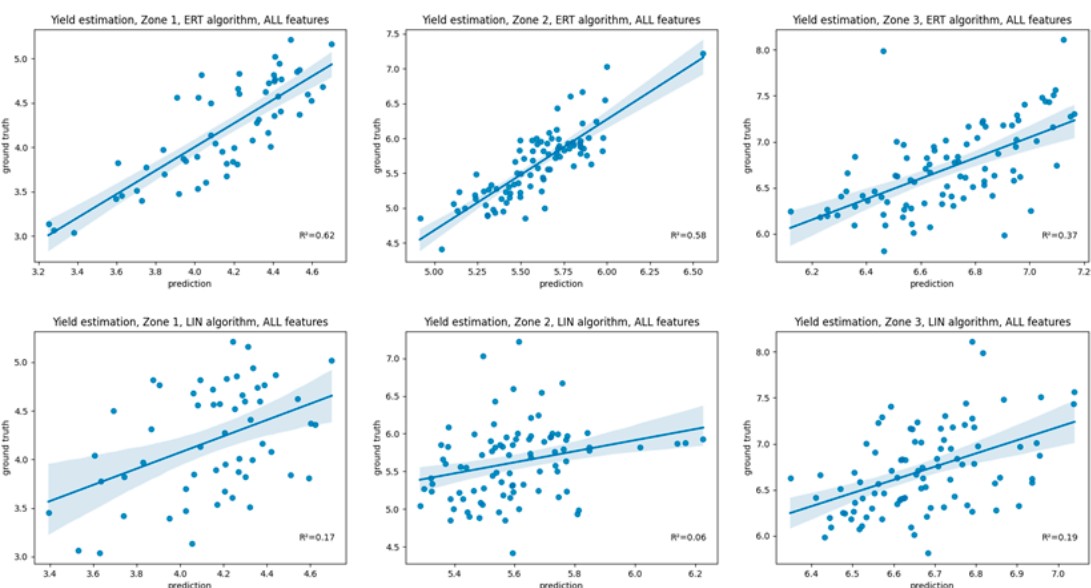

**Figure 8.** Validation for the generic yield estimation by 3 productivity levels with ERT (**upper row**) and linear regression (**lower row**).

### 4.2.3. Generic Yield Model

This experiment aimed to produce a generic yield model, by sampling through all crop types and productivity levels, and to prove whether the inclusion of CCM-based features achieves higher $R^2$ scores than NDVI-only models. Thus, the experiment tested feature sets of CCM features (achieving $R^2$ of 0.84), NDVI features (with $R^2$ of 0.88), and the fusion of both feature sets by PCA transform (where the common explained variance is in the first components), achieving $R^2$ of 0.92 with Random Forest regression and $R^2$ of 0.41 with linear regression (whereas the linear model fails when using only NDVI or CCM features).

As shown in Figure 9, the contribution of additional CCM features to the NDVI dataset does not improve the accuracy of the ERT model, and the use of NDVI (or possibly any other spectral vegetation index) is considered sufficient for the analysis.

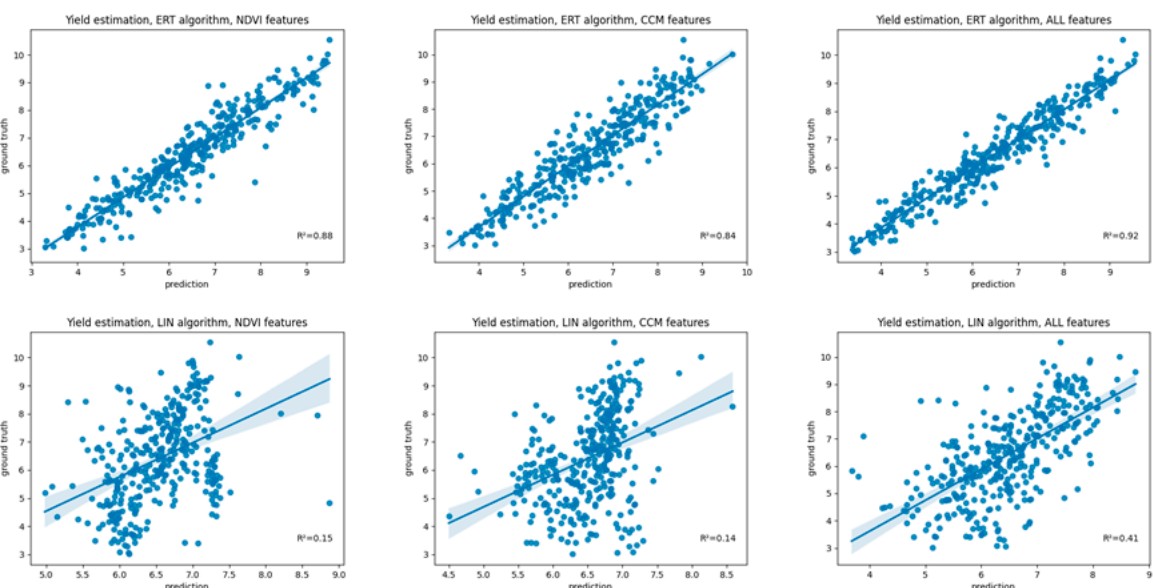

**Figure 9.** Validation results for a generic yield estimation with different sets of features (CCM, NDVI, and PCA-transformed fusion) demonstrated the better performance of the ERT algorithm (**upper row**) over the linear regression (**lower row**).

### 4.2.4. Stratified Cross-Validation (Crop-Agnostic Transfer)

The stratified cross-validation proves whether the model is crop-agnostic and whether the features can be utilized arbitrarily, assuming that the multi-temporal NDVI and CCM-based features contribute to crop-independent yield estimation. The results shown in Figure 10 (depicting the different $R^2$ scores) reveal that some yield estimation models trained on a specific crop type are transferable to other crop types (winter wheat to winter oats and canola, triticale to winter wheat). In contrast, in other cases, the transferability fails for certain crop types (winter oats, winter wheat, triticale or canola to winter barley, winter wheat to triticale). The lack of responsive soil data (e.g., cation exchange capacity, clay content, or soil organic carbon) or management data (e.g., plowing, fertilization) could explain the model failure.

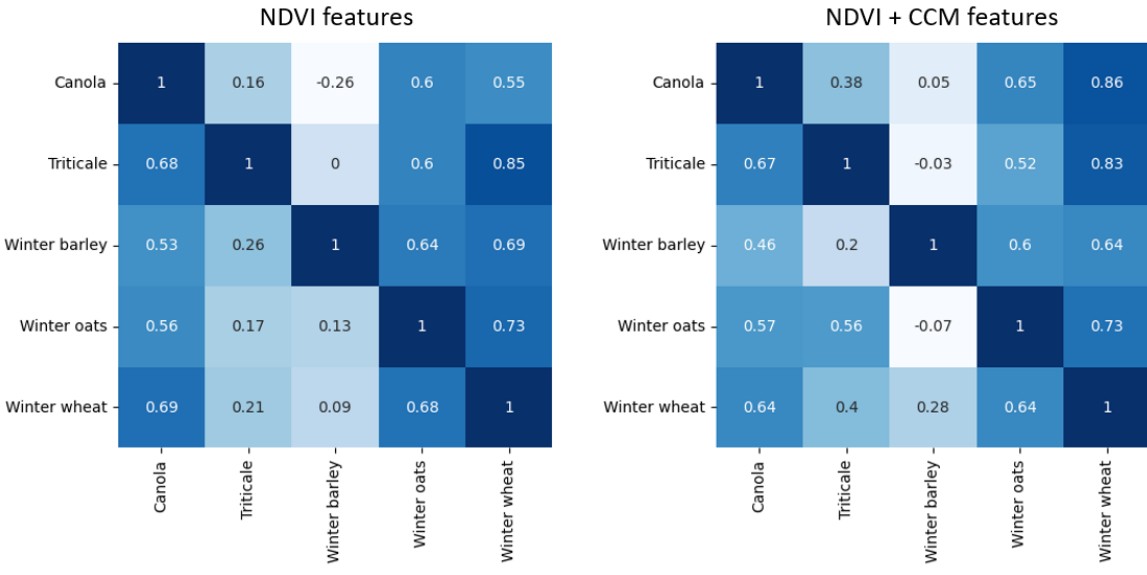

**Figure 10.** Cross–validation results ($R^2$ scores) of the yield estimation by different crop types based on all NDVI features (**left**) and NDVI and CCM features (**right**).

Our analysis proves that the inclusion of CCM-based features for the model transfer improves estimation accuracies for some crop types (canola to winter wheat, winter oats to triticale, and winter wheat to winter barley). In contrast, no improvement could be measured (canola to winter barley, winter oats to canola), or even a decrease in model robustness took place, through assumed contamination of the predictor variables (winter oats to winter barley, triticale to winter oats).

### 4.2.5. Multitemporal vs. Single Observation

We analyzed the feature importance of both datasets to assess the necessity of multitemporal observations (whose acquisition would be much more expensive). For each feature and observation date, we conducted a univariate regression which did not achieve a higher $R^2$ score than 0.2 in all cases. We assume this is due to processing errors (gaps, spikes, view angle shadows, mismatching) within the CCM, and it is unmistakable evidence that the yield estimation requires multi-temporal data that captures the phenology and considers the crop development over time.

Figure 11 depicts the importance of different observation dates for both datasets, showing that the most significant dates are those close to the peak of the season (or maximum biomass), whereas the dates at or after harvest (with the satellite data capturing harvest residuals and bare soil) have a much lower impact on the model. This is shown through SHAP metrics (SHapley Additive exPlanations), which not only show the importance of each feature but also the positive (positive *x*-axis) or negative (negative *x*-axis) impact on the model with lower (red) or higher (blue) feature values, respectively, as well as the distribution of the feature values [29].

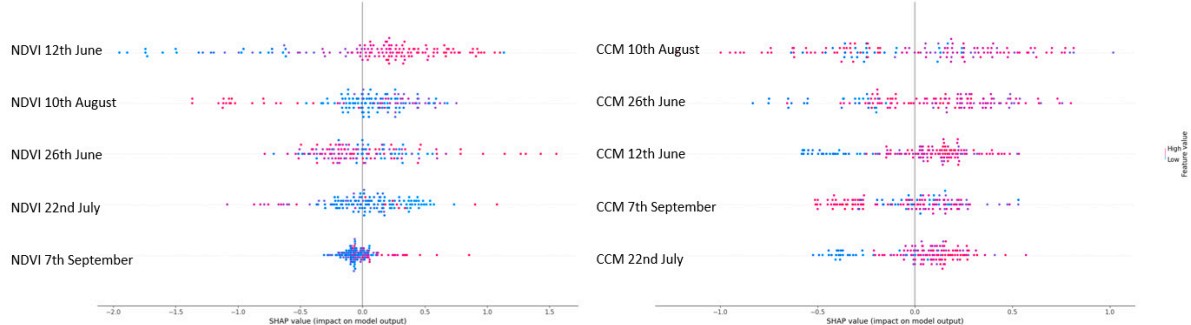

**Figure 11.** SHAP metrics for the five different observation dates, and the NDVI (**left**) and CCM (**right**) features with low (red) and high (blue) feature values and their respective negative and positive contributions to the model.

SHAP metrics can be used as an alternative feature selection method to the natively implemented feature importance metrics of tree-based estimators (Random Forest, Gradient Boosting Trees, Adaptive Boosting, and the ERT algorithm used in this study). Features with a very low model contribution can be neglected for further use, such as the last NDVI observation from 7 September, as seen in Figure 11.

### 4.3. Assessment of Growth Characteristics

In all cases, except the field "Theateracker" (with winter oats), the nDSM values follow the decreasing NDVI values at the season's end for all crops (July–August). This observation confirms the initial assumption that, while different crops are maturing after maximum greenness (maximum biomass at the peak of the season), the chlorophyll content is decreasing (yellowing, ripening stages) and leading to a delayed decrease in crop stem heights (pendent stem heads) [30,31]. Moreover, as shown in Figure 12, harvest activity also leads to reduced crop height and decreased NDVI levels by removing stems and above-ground biomass from the respective field.

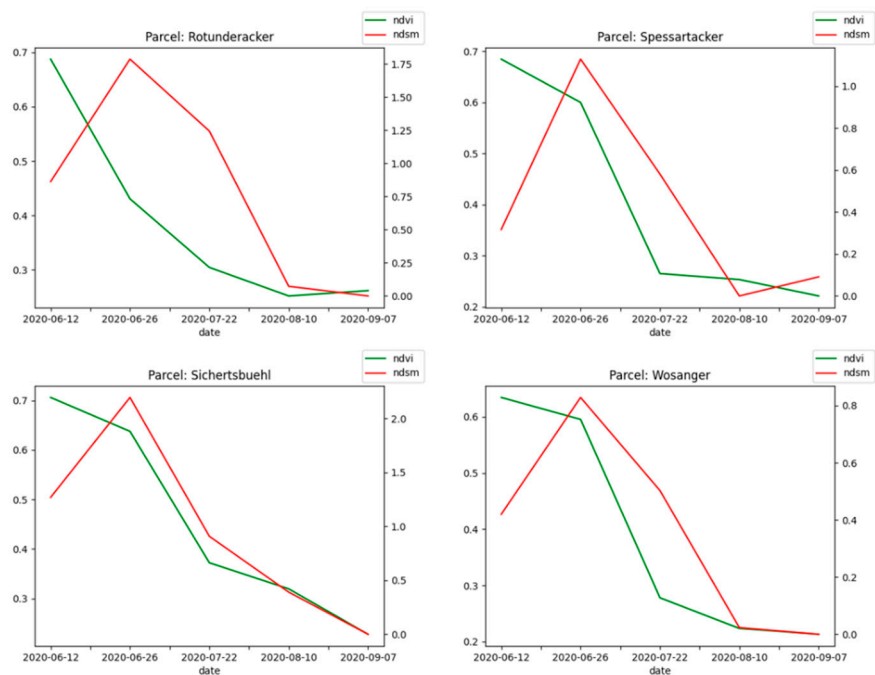

**Figure 12.** Multi-temporal depiction of the aggregated NDVI and nDSM features for 4 of the test parcels.

## 5. Conclusions

Generating DSMs from Pleiades imagery without pre-defined GCPs and relying only on relative orientation through tie points is possible, but it needs to be more accurate to derive absolute (normalized) heights. The computed DSMs, however, capture vertical vegetation changes and can be used as an additional data source to reveal crop growth dynamics along with NDVI time series.

While the derivation of normalized nDSM from computed DEMs may be sufficient for forestry and urban mapping use cases, the omission of essential terrain features (through the required filtering of gaps and spikes with too abrupt slope changes) worsens the extraction of exact crop heights, especially for crops with a height less than 2 m. Our results have shown that bi-stereo imaging is not sufficient for those purposes.

Future model improvements would include tri-stereo instead of bi-stereo Pleiades data, precisely defined GCPs, and using a pre-computed reference DEM to extract normalized heights. Moreover, deep learning methods may improve the results by convolutional filtering as implemented in Convolutional Neural Networks—mainly to contextualize the spatial relationship of horizontal and vertical structures expressed by NDVI and DSM features, respectively.

Pleiades 1a/1b data are feasible for high-resolution yield estimation through NDVI features. Further research would include the utilization of other spectral indices, which can be derived from the Pleiades 1a/1b sensor, such as the Soil-Adjusted Vegetation Index (SAVI), the Enhanced Vegetation Index (EVI), or the Green Chlorophyll Index (GCI). However, including additional CCM features has yet to improve the generic (crop-agnostic) yield estimation, where the NDVI is the most contributive variable. On the other side, the crop-specific yield estimation and the transfer modeling between different crop types showed that including CCM features stabilizes predictions and decreases model errors, where crop height seems essential for yield estimation. However, the costs of stereo data are high; on average, depending on the tasking mode (single image or multiple image series acquisition) and the national data distributor, the price can vary between EUR 28 and EUR 38 per km$^2$ with a minimum area of interest between about 100 km$^2$ and 250 km$^2$.

Moreover, the processing costs for the photogrammetric modeling and the resulting gains in yield estimation accuracy need to be evaluated and compared against the costs, drawbacks, and benefits (e.g., spatial resolution, acquisition swath, timeliness of data

provision) of the utilization of UAV [32]. Especially for crop types where the biomass (represented in the volumetric CCM features) or crop height is not related to the yield, such as legumes, tubers, orchards, or vegetable crops, we do not recommend the inclusion of CCMs as input features. In addition, the operational use of open-source software for deriving a DSM from stereo imagery or bundle adjustment is still not mature enough, and the usage of expensive proprietary software might be necessary.

**Author Contributions:** Both authors carried out the conceptualization, methodology and validation in a joint work effort. All authors have read and agreed to the published version of the manuscript.

**Funding:** This study is self-funded, and the acquired Pleiades-1a data free for research purposes.

**Data Availability Statement:** The data and results that support the findings of this study are available from the corresponding author upon reasonable request.

**Acknowledgments:** Special thanks to the Airbus Defence and Space and Spot Image France team for the great support and provision of the free Pleiades stereo data for our research. Thanks also to Livia Piermattei for scientific consultancy on photogrammetric modeling and to Markus Heinz from the *Landwirtschaftliche Lehranstalten Triesdorf* for providing the yield data.

**Conflicts of Interest:** The authors declare that they have no known competing financial interest or personal relationships that could have appeared to influence the work reported in this paper.

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
