# Peer review of "Exploring the Potential of Multi-Temporal Crop Canopy Models and Vegetation Indices from Pleiades Imagery for Yield Estimation"

_remotesensing, doi:10.3390/rs15163990_

Round 1

Reviewer 1 Report

My comments on the manuscript include,

(1) The current edits are more close a report rather than a scientific paper. The main issue is that in-depth assessments and discussions on the yield modeling using the parameters extracted from Pleiades imagery are missing in the current version.

(2) R2 was the only statistical indicator used for assessing and validating the yield modeling. However, the accuracy of yield estimation may be very bad for a case that a very strong linear relationship between estimated and measured yield exists but having a slope value far from 1.

(3) In-depth explanations and discussions on  the SHAP metrics (figure 11) for the contributions of parameters from Pleiades data on yield estimation, especially the physical-based background on the yield modeling using VIs or CCM, are needed.

(4) Line 355 - 358, how did the authors know chlorophyll content playing an important role on this change. Supporting documents and data on this conclusion are needed.

Reviewer 2 Report

The experimental topic is a current and important area. The importance and usefulness of data collected by remote sensing is constantly increasing.

A more extensive presentation of the soil parameters is necessary. Is there heterogeneity in the given area, and if so, which soil parameters does it cover?

Needed the ASL near to the coordinate. = Altitude from the sea level.

How was the amount of the yield measured? Please discribe the yield mapping method, and the calibration if there was one. 

The article is well edited, but the experimental methodology, mainly the basic agrotechnical and plant breeding harvesting information, needs to be supplemented.

The experiment collects data from a small area, the geospatial analysis of which in many cases is more appropriate to be done with a drone, especially in the case of DSM and DTM models. It results in more accurate and reliable values, especially when using RTK. If such data is available, it would be advisable to add it to the manuscript. 

The quality of the article could be improved by editing the figures and tables in a better, more aesthetic way, if possible.

Reviewer 3 Report

The manuscript entitled "Exploring the potential of multi-temporal crop canopy models and vegetation indices from Pleiades imagery for yield estimation" is bringing the crucial information on some limitations in using sensors as technology to estimate crop yields and what can be done to reduce these limitations. The research findings will be beneficial to the famers especially smallholders, researchers, agro-industries, students, other readers, etc. I suggest the following to improve the quality of the manuscript:

1. Abstract is lacking the clear purpose and methods of the study. Authors need to attend to this.

2. Authors must follow MDPI guidelines when citing references in the Text. The Reference number must be placed in the Text instead of the Author (s) and this number must be in square brackets (refer to Instructions for Authors). This must be done throughout the manuscript.

3. Authors must indicate the objective of this research study and also the hypothesis of the study clearly towards the end of the Introduction Section.

4. Authors must write Materials and Methods instead of Data and methods. The subsection must be Study area and Data collection in Line 111

5. In the sentence that appear in Line 122, the authors must add "with an area ranging from 1.08 ha to 4.53" immediately after part of the sentence written "on eight fields" and then continue with the sentence as is. The mentioning of the size of the area will be beneficial to the readers especially to those who still want to use remote sensing agricultural technologies for the first time but are not sure of the size of the area where this technology can be used especially the smallholder farmers.

6. Authors must relook and read the sentence starting from Line 163 to Line 166 because something is missing in it and authors need to correct this.

7. In Line 167, authors must replace subsection Methodology with Data Analysis.

8. The manuscript lack discussions of the results and the authors must attend to this. The Discussion section is important because it is where the authors give interpretation of the  findings of study in relation to the previous studies and also the implications and limitations of the study. 

9. The authors need to relook at the Conclusion Section because some information under Conclusion can be part of the Discussion.

10. Listing of References must be in accordance with the order of reference citation in the text.

11. The manuscript also lack the following: Funding; Author Contributions; Conflict of Interests; Data availability statement and the authors need to attend to this.

The Quality of English Language is fine but needs minor editing.

Reviewer 4 Report

Reviewer's Comments

 Title:  Exploring the potential of multi-temporal Crop Canopy Models and Vegetation Indices from Pleiades imagery for yield estimation

  Manuscript ID: remotesensing-2485193

General Remarks

Generally, the article remotesensing-2485193 entitled " Exploring the potential of multi-temporal Crop Canopy Models and Vegetation Indices from Pleiades imagery for yield estimation" demonstrates one of the widely used technique (analysis of remotely sensed data) to explore different CCM normalization methods, linear and non-linear regression algorithms for yield estimation. The manuscript is clearly written, with some Minor comments.

1-      There  are numerous language issues (sentence structure, grammatical errors, and typos) in this manuscript. The entire document has to have its language edited carefully.

2-      In the title of manuscript insert the country of studied area.

3-      The figures need to more quality in all manuscript

Round 2

Reviewer 3 Report

The author has addressed my comments.

The English language is fine but needs minor editing.